# Nuclear myosin VI maintains replication fork stability

Jie Shi[1], Kristine Hauschulte [1], Ivan Mikicic [1], Srijana Maharjan[1,4], Valerie Arz[1], Tina Strauch[1], Jan B. Heidelberger[1,5], Jonas V. Schaefer [2], Birgit Dreier[2], Andreas Plückthun [2], Petra Beli [1,3], Helle D. Ulrich [1] ✉ & Hans-Peter Wollscheid [1] ✉

The actin cytoskeleton is of fundamental importance for cellular structure and plasticity. However, abundance and function of filamentous actin in the nucleus are still controversial. Here we show that the actin-based molecular motor myosin VI contributes to the stabilization of stalled or reversed replication forks. In response to DNA replication stress, myosin VI associates with stalled replication intermediates and cooperates with the AAA ATPase Werner helicase interacting protein 1 (WRNIP1) in protecting these structures from DNA2-mediated nucleolytic attack. Using functionalized affinity probes to manipulate myosin VI levels in a compartment-specific manner, we provide evidence for the direct involvement of myosin VI in the nucleus and against a contribution of the abundant cytoplasmic pool during the replication stress response.

Complete and correct duplication of the genome in each cell cycle is crucial for genome stability in proliferating cells. One of the many protective responses to DNA replication stress is the reversal of replication forks, involving a reannealing of the parental strands and a joining of the newly synthesized strands into a four-way Holliday junction-like structure[1,2]. However, fork reversal, mediated by DNA-remodeling factors such as RAD51, SMARCAL1, HLTF, and ZRANB3[3–5], can also be detrimental for genome stability. Due to their structure resembling a one-ended double strand break (DSB), reversed forks can become targets of nucleolytic attack by nucleases such as DNA2 and MRE11, resulting in fork instability and collapse[6].

The actin cytoskeleton exerts a fundamental role in cell mechanics, motility and intracellular transport. Filamentous (F−) actin is highly abundant in the cytoplasm but barely detectable in the nucleus, where its functional relevance is still controversially discussed[7,8]. Recent discoveries have connected nuclear F-actin to genome maintenance pathways such as DSB repair, DNA replication and maintenance of nuclear architecture[9–14]. If and how myosins in their function as actin-based molecular motor proteins participate in these processes is still poorly understood. The myosin superfamily comprises more than 35 distinct classes, of which only a few have been shown to exert nuclear functions in humans[15]. Based on their presumably higher degree of functional specialization compared to actin, investigation of myosins rather than actin itself provides a unique opportunity to tease apart specific functional aspects of actin-mediated dynamic processes while avoiding the pleiotropic effects of manipulating actin directly.

Myosin VI, the only minus end-directed myosin characterized to date[16], is well known for its contribution to multiple steps of the transcriptional process[17–19]. Here we report an association of myosin VI with numerous components of the replication machinery. Upon replication stress, we found myosin VI to cooperate with Werner helicase interacting protein 1 (WRNIP1) in the protection of stressed replication forks from DNA2-mediated degradation.

## Results

### Myosin VI interacts with replisome components and protects reversed forks from nuclease-mediated degradation

We recently identified a region adjacent to the C-terminal cargo-binding domain of myosin VI as a ubiquitin-interacting domain

[1]Institute of Molecular Biology gGmbH (IMB), Ackermannweg 4, D – 55128 Mainz, Germany. [2]University of Zurich, Department of Biochemistry, Winterthurerstr. 190, CH – 8057 Zurich, Switzerland. [3]Institute of Developmental Biology and Neurobiology, Johannes Gutenberg University, Hanns-Dieter-Hüsch-Weg 15, D – 55128 Mainz, Germany. [4]Present address: Mainz Biomed N.V., Robert-Koch-Str. 50, D – 55129 Mainz, Germany. [5]Present address: Max Planck School Matter to Life, Jahnstr. 29, D – 69120 Heidelberg, Germany. ✉e-mail: h.ulrich@imb-mainz.de; h.wollscheid@imb-mainz.de

(MyUb, Fig. 1a)[20]. Pulldown assays with a GST-MyUb construct, followed by SILAC-based quantitative mass spectrometry (Fig. 1b), identified 490 proteins with an at least twofold enrichment over the GST control (FDR < 0.05; Supplementary Data 1, Supplementary Fig. 1b), including 346 proteins annotated with the gene ontology (GO) cellular

compartment "nucleus" (Supplementary Data 2). In line with its known function, GO term analysis of the MyUb interactome showed transcription-associated proteins as the most prominently enriched category (Fig. 1c, Supplementary Data 3). We also detected GIPC1, a well-described cytoplasmic myosin VI cargo (Fig. 1d). In addition, we

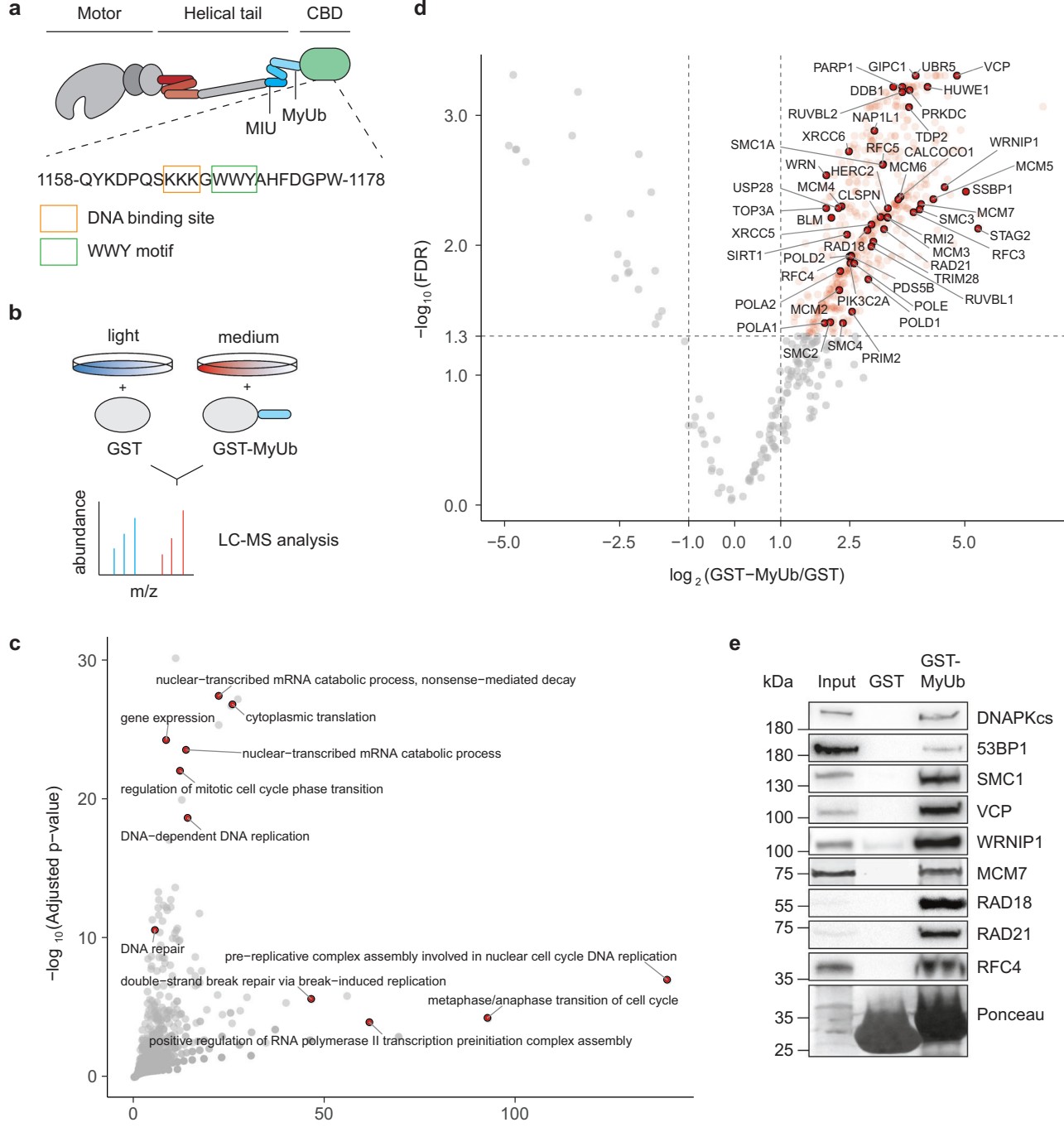

**Fig. 1 | Myosin VI interacts with the replisome. a** Schematic representation of myosin VI (adapted from Magistrati and Polo[40]) showing the positions of the ubiquitin-binding MIU and MyUb domains (blue) adjacent to the cargo-binding domain (CBD, green). The amino acid sequence shows a triple-Lys repeat involved in DNA binding[18] (orange box) and the WWY motif (green box), a well-characterized protein interaction site. The three-helix bundle at the N-terminal tail is indicated in red. Amino acid numbering is according to the short isoform (isoform 2). **b** Set-up of the SILAC experiment for identification of MyUb interaction partners. **c** GO term analysis (GO biological process) of proteins identified to interact with the MyUb

domain (fold change > 4, FDR < 0.05) using EnrichR. **d** Volcano plot of protein groups identified in the SILAC interactome experiment. Mean $\log_2$ fold change of all replicates between GST-MyUb and GST are plotted against the $-\log_{10}$ FDR. Significantly enriched proteins are shown in red (fold change > 2, FDR < 0.05). Interactors involved in DNA replication and repair are highlighted and labeled. **e** Validation of selected candidates by pulldown assays from total cell lysates with recombinant GST-MyUb, followed by western blotting and Ponceau S staining. Results were confirmed by at least two independent experiments. Source data are provided as a Source Data file.

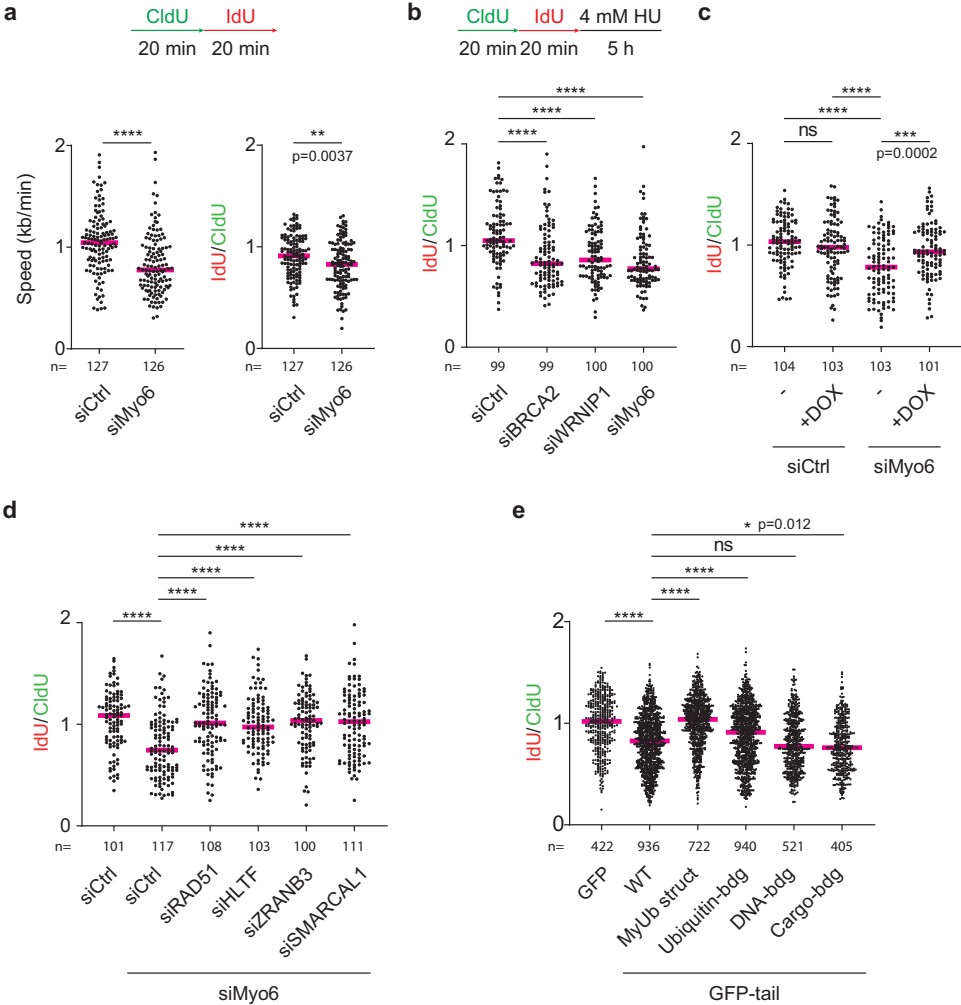

**Fig. 2 | Myosin VI protects stalled replication forks from degradation. a** Myosin VI is required for efficient unperturbed DNA replication. Top: Schematic representation of fiber assay conditions. Bottom: Fiber assays performed on siRNA-transfected U2OS cells. Left panel: replication speed, measured as total track lengths (CldU+IdU). Right panel: IdU/CldU ratios. **b** Depletion of myosin VI via siRNA causes erosion of stalled replication forks. Top: Schematic representation of fiber assay conditions. Bottom: Fiber assays performed on siRNA-transfected U2OS cells. **c** GFP-myosin VI complements the loss of endogenous myosin VI. U2OS cells harboring DOX-inducible GFP-myosin VI were siRNA-transfected (siRNA #10 targeting myosin VI) and treated −/+ 20 ng/ml DOX for 24 h, followed by fiber assays as shown in (**b**). **d** Myosin VI-dependent fork protection requires fork reversal factors. U2OS cells were siRNA-transfected as indicated, followed by fiber assays as shown in (**b**). **e** Motor- and MyUb-domains of myosin VI are required for fork protection. GFP-tail *wildtype* (WT) and mutants were overexpressed in U2OS cells, followed by fiber assays as shown in (**b**). Combined data from at least three independent replicates are shown. Detailed information about the respective mutations is given in Supplementary Fig. 2f. For all panels: IdU/CldU ratios are shown as dot plots with median values (red bars). Significance levels were calculated using the two-sided Mann–Whitney test from the indicated number of fibers per sample (ns: not significant, ****: $p < 0.0001$, ***: $p < 0.001$, **:$p < 0.01$, *: $p < 0.05$) and annotated for $p > 0.0001$. Knockdown efficiencies and overexpression levels are shown in Supplementary Fig. 2. For (**a–d**): A representative experiment from three independent replicates is shown. Source data are provided as a Source Data file.

identified many DNA replication-associated factors, suggesting a yet unidentified function of myosin VI at the replisome (Fig. 1c, d). Immunoprecipitation (IP) experiments upon overexpression of GFP-tagged putative interactors (Supplementary Fig. 1c) or pulldown experiments with recombinant GST-MyUb followed by immunoblotting with antibodies against endogenous proteins (Fig. 1e) validated many of the candidates identified in our proteomic screen as genuine interaction partners of myosin VI.

To assess a potential role of myosin VI during DNA replication, we measured replication speed using DNA fiber assays, where nascent DNA is labeled consecutively with two thymidine analogues, CldU and IdU. Knockdown of myosin VI did not lead to detectable changes in the activation pattern of the checkpoint kinase ATR in the absence or presence of replication stress (Supplementary Fig. 2b) but caused a reduction in overall unperturbed DNA replication speed, suggesting its requirement for efficient DNA replication (Fig. 2a, left panel). To

determine whether this reduction was attributable to an overall slowing of replication fork progression or rather an increase in the frequency of fork breakdown, we calculated the IdU/CldU ratio as an estimate of the extent to which forks irreversibly stall during the IdU pulse. A reduction of this value upon myosin VI knockdown suggested an increased propensity of fork stalling or possibly a defect in fork recovery after stalling (Fig. 2a, right panel).

The AAA-ATPase WRNIP1 has been implicated in genome maintenance as a protector of reversed replication forks[21,22]. Considering its identification as an interaction partner of myosin VI (Fig. 1d, e, Supplementary Fig. 1c), we asked whether the replication problems upon myosin VI depletion were linked to a defect in the protection of stalled forks. To distinguish fork degradation from fork stalling, we labeled cells with CldU and IdU for 20 min each, followed by a 5 h treatment with hydroxyurea (HU) (Fig. 2b). In this setup, any additional shortening of the IdU tract is an indication of nascent strand degradation

during the HU treatment. According to their well-established roles as replication fork protectors, siRNA-mediated depletion of WRNIP1 and BRCA2[23] resulted in a reduction in the IdU/CldU ratio (Fig. 2b). Notably, myosin VI depletion reduced this ratio to a similar extent, suggesting that myosin VI is essential for preventing nuclease-mediated degradation of reversed forks (Fig. 2b). To exclude off-target effects, we carried out rescue experiments using a cell line expressing siRNA-resistant GFP-myosin VI under the control of a doxycycline (DOX)-inducible promoter (Fig. 2c, Supplementary Fig. 2d). In control cells expressing endogenous myosin VI, addition of DOX did not significantly alter the stability of stalled replication forks (Fig. 2c, lanes 1 and 2). However, in myosin VI-depleted cells, we observed a rescue of fork protection upon DOX-induced restoration of myosin VI levels (Fig. 2c, lanes 3 and 4), thus verifying the direct correlation between replication fork stability and myosin VI abundance. Furthermore, co-depletion of the fork remodelers RAD51, HLTF, SMARCAL1 or ZRANB3 together with myosin VI completely abolished nascent strand degradation (Fig. 2d), indicating that the defect in fork stability induced by myosin VI depletion depends on the prior action of the fork remodelers. Thus, myosin VI appears to protect reversed replication forks, but it does not prevent fork reversal.

To elucidate the molecular characteristics of myosin VI-dependent fork protection, we made use of a motor-deficient variant (GFP-tail, Supplementary Fig. 2f). Its overexpression resulted in nascent strand degradation similar to myosin VI depletion (Fig. 2e), demonstrating the importance of its motor activity for fork protection. By exploiting this dominant-negative effect, we addressed the contributions of multiple functional domains of myosin VI (Supplementary Fig. 2f, g) to the replication stress response. It was previously shown that mutation of the RRL motif within the MyUb domain leads to destabilization of its helical structure[20]. In line with the multitude of replication factors that interact with this domain, mutation of the RRL motif to AAA abolished the dominant-negative effect of the GFP-tail construct (Fig. 2e, lane 3). A combination of point mutations in the MIU (A1013G)[24] and MyUb (I1072A)[20] domains revealed a contribution of ubiquitin binding to myosin VI´s activity in fork protection, whereas its DNA[18]- and WWY[25]-mediated cargo-binding activities (Fig. 1a, Supplementary Fig. 2f) seem to be less important (Fig. 2e).

## Myosin VI cooperates with WRNIP1 to protect stalled forks from DNA2-mediated degradation

In contrast to other fork protectors, myosin VI primarily localizes to the cytoplasm. Even upon replication stress, where nuclear actin and F-actin levels slightly increase, we did not observe an accumulation of myosin VI in the nucleus (Supplementary Fig. 3a–f). To investigate a potential physical association with ongoing and stalled or reversed replication forks, we therefore utilized iPOND (isolation of proteins on nascent DNA) with western blotting to focus specifically on chromatin-associated factors[26]. PCNA is known to dissociate from newly replicated DNA upon replication stress[27], and this pattern was reproducible in our hands (Fig. 3a). In agreement with the observed interactions of myosin VI with replisome components (Fig. 1c, e), we detected myosin VI at unperturbed replication forks (Fig. 3a). Unlike PCNA, however, myosin VI association was not diminished upon HU treatment. To achieve a more quantitative assessment, we used SIRF (in situ protein interaction with nascent DNA replication forks) assays, which detect the co-localization of a protein of interest with nascent, EdU-labeled DNA via proximity ligation[28]. Again, the PCNA signal was lost under conditions of replication stress, while both myosin VI and WRNIP1 showed enhanced association with EdU-positive nascent DNA upon HU treatment (Fig. 3b), suggesting an enrichment of both myosin VI and WRNIP1 at stalled forks.

Having established the interaction of WRNIP1 with the MyUb domain of myosin VI (Fig. 1e, Supplementary Fig. 1c), we utilized proximity ligation assays (PLA) to validate this interaction in living cells using antibodies against the endogenous proteins (Fig. 3c). Strikingly, the PLA signal was prominently enhanced under conditions of replication stress, suggesting that the proteins preferentially interact at stalled replication forks (Fig. 3c). To assess whether the interaction of WRNIP1 with myosin VI is direct or possibly mediated via common association on ubiquitin conjugates through their respective ubiquitin-binding domains[20,29], we performed GST-pulldown assays with bacterially expressed recombinant proteins. We detected a direct interaction of a His-tagged MIUMyUb domain construct with GST-WRNIP1 (Fig. 3d) that was further enhanced by the addition of K63-linked polyubiquitin chains (Fig. 3e), indicating a potential modulation of the myosin VI-WRNIP1 interaction by ubiquitin signaling. To verify this effect with endogenous proteins, we pre-treated cellular lysates with a non-selective de-ubiquitylating enzyme, His-USP2cc[30], resulting in the disassembly of endogenous ubiquitin conjugates. A significant decrease in WRNIP1 binding to myosin VI was detected upon His-USP2cc treatment, further supporting the relevance of polyubiquitin chains for myosin VI´s association with WRNIP1 (Fig. 3f).

Unlike BRCA2, which is thought to protect the ends of the regressed arm from MRE11-dependent degradation[23], WRNIP1 was reported to prevent attack by SLX4/DNA2 at the four-way junction[21] (Fig. 3g). To specify the nature of myosin VI activity at reversed forks, we performed DNA fiber assays in the presence of the MRE11- or DNA2-specific inhibitors mirin or C5, respectively. Consistent with previous findings[21], mirin treatment did not rescue nascent strand degradation in WRNIP1-depleted cells, while DNA2 inhibition led to a full stabilization of reversed forks (Fig. 3h). Use of the inhibitors in myosin VI-depleted cells resulted in a very similar pattern (Fig. 3h), suggesting that myosin VI cooperates with WRNIP1 to protect reversed replication forks from DNA2-mediated nucleolytic attack.

## A functionalized DARPin verifies the contribution of myosin VI to fork stabilization

Actin filaments are of a transient nature and difficult to detect in the nucleus because of their high cytoplasmic abundance. An actin-specific nanobody fused to a nuclear localization signal (NLS), termed nuclear actin chromobody (nAC), has proven to be a valuable instrument in visualizing nuclear F-actin specifically[31]. However, manipulation of nuclear F-actin remains challenging due to the involvement of monomeric actin in chromatin remodeling complexes[32] and its association with RNA polymerase complexes[33–35]. Inspired by the nAC technology, we aimed to develop tools to manipulate the stability and localization of endogenous myosin VI. To obtain a myosin VI-specific affinity probe, we employed a ribosome display library of designed ankyrin repeat proteins (DARPins)[36], which consist of stacked repeat modules with a randomized surface. They can be selected to bind proteins with antibody-like selectivity and affinity[36–38]. Unlike antibodies, DARPins fold under the reducing conditions of the cytoplasm and the nucleus and can thus be expressed and folded in these compartments. After selection of DARPins using a biotinylated tail fragment of myosin VI (aa 992 – 1122) as bait, the enriched pool was subcloned into an E. coli expression vector and crude bacterial extracts of 380 individual clones were tested for DARPin binding to the biotinylated fragment of myosin VI in HTRF assays (Supplementary Fig. 4a). Among these, 54 high-scoring clones (>30% signal over background) and 46 clones with low signal intensity (>5%) were identified as initial hits. From these, 52 random clones were chosen and further screened in GST-pulldown experiments (examples are shown in Supplementary Fig. 4b). Five positive clones were tested for their ability to deplete endogenous myosin VI from cellular lysates, using a non-binding DARPin (E3_5)[38] as negative control. One clone, M6G4, effectively depleted myosin VI from the lysate (Fig. 4a, Supplementary Fig. 4d, Supplementary Fig. 7) and was therefore selected as the target-binding module for the myosin VI-specific tools. Using Surface Plasmon Resonance (SPR) assays, we measured a dissociation constant for

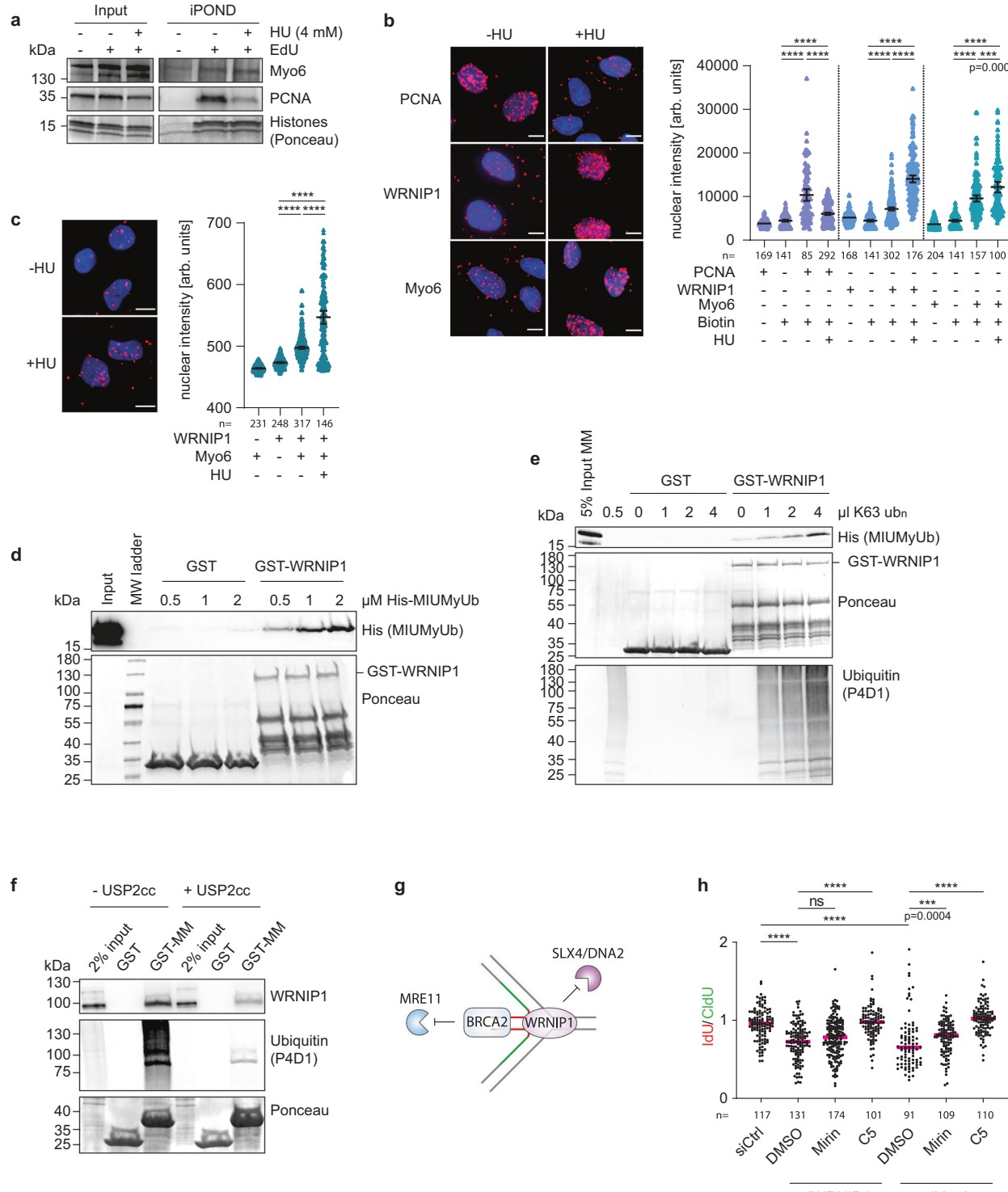

M6G4 of ca. 60 nM (Supplementary Fig. 4e). Notably, M6G4 does not interfere with the binding of endogenous interactors like WRNIP1 or ubiquitin to myosin VI (Supplementary Fig. 4f).

To generate a myosin VI-specific degradation tool, we adapted a recently published system based on the ubiquitin protein ligase RNF4[39] (Supplementary Fig. 5a). A fusion construct of DARPin M6G4 with two RING finger domains of RNF4 (M6G4-2RING) was stably integrated in the chromosomes under the control of a DOX-inducible promoter. A single-cell clone termed 2R#8 showed efficient proteasome-dependent degradation of endogenous myosin VI in a time- and DOX-dependent manner (Fig. 4b, Supplementary Fig. 5b–d).

Importantly, depletion of myosin VI via M6G4-2RING resulted in a destabilization of stalled forks, comparable to siRNA-mediated myosin VI depletion (Fig. 4c), providing additional support for the specificity of the phenotype.

## The nuclear but not the cytoplasmic pool of myosin VI contributes to fork protection

Having verified the selectivity of the M6G4 probe, we asked whether fork stability was regulated by the nuclear or the cytoplasmic pool of myosin VI. We found that inducible expression of a GFP-tagged fusion construct of M6G4 to a 3xNLS resulted in a nearly complete

**Fig. 3 | Myosin VI cooperates with WRNIP1 to protect stalled forks from DNA2-mediated degradation. a** iPOND assays show localization of myosin VI at replication forks. U2OS cells, 30 min EdU-pulsed −/+ 4 mM HU. Chromatin-bound proteins are visualized using western blotting and Ponceau S staining. **b** SIRF assays confirm the presence of myosin VI at replication forks. U2OS cells, 30 min EdU-pulsed −/+ 4 mM HU, followed by click reaction with Biotin azide and standard PLA assay. **c** Interaction of myosin VI with WRNIP1 is enhanced upon replication stress. U2OS cells, −/+ 4 mM HU, followed by standard PLA assay. For (**b**, **c**): Left: representative images, Hoechst (blue), PLA (magenta), scale bar = 10 μm. Right: dot plots of PLA signal intensities with mean values −/+ 95% confidence intervals. **d** WRNIP1 interacts directly with the MIUMyUb domains of myosin VI. GST-pulldown assay with recombinant proteins, visualized by western blotting and Ponceau S staining. Input represents 5% of the 2 μM MIUMyUb sample. **e** K63-linked ubiquitin chains enhance the WRNIP1-myosin VI interaction. GST-pulldown as in (**d**), using 1 μM His-MIUMyUb and increasing concentrations of K63 poly-ubiquitin. **f** Depletion of ubiquitin conjugates interferes with the WRNIP1-myosin VI interaction. GST-pulldown assay with recombinant baits and cellular lysates, pre-treated for 10 min at 37 °C −/+ 5 μM USP2cc. Visualization by western blotting and Ponceau S staining. **g** Schematic representation of fork protection mechanisms by WRNIP1 and BRCA2 according to Porebski et al.[21] with nascent DNA colored in green and red. **h** Inhibition of DNA2 restores fork stability in WRNIP1- and myosin VI-deficient cells. Fiber assays performed on siRNA- transfected U2OS cells, −/+5 h nuclease inhibitor treatment, as in Fig. 2b. IdU/CldU ratios are shown as dot plots with median values. Knockdown efficiencies are shown in Supplementary Fig. 3g. For (**a**–**f**, **h**): A representative experiment from three independent replicates is shown. For (**b**, **c**, **h**): Significance levels were calculated using the two-tailed Mann–Whitney test from the indicated number of nuclei or fibers per sample (ns: not significant, ****: $p < 0.0001$, ***: $p < 0.001$) and annotated for $p > 0.0001$. Source data are provided as a Source Data file.

localization of myosin VI to the nuclear compartment (Fig. 4d), while the analogous GFP-NLS-E3_5 control construct (with a non-binding DARPin) did not afford significant changes in the subcellular distribution of myosin VI. Fiber assays in cells expressing either the myosin VI-specific or the control NLS-DARPin did not show significant degradation of newly replicated DNA (Fig. 4e), suggesting that depletion of cytoplasmic myosin VI has little or no influence on fork stability. Unfortunately, our attempts to selectively deplete myosin VI from the nucleus by fusion of an analogous nuclear export signal (NES) were inconclusive due to low expression of the NES-M6G4 construct and difficulties visualizing the nuclear pool of myosin VI.

As an alternative approach, we therefore expressed motor-deficient myosin VI mutants (NLS/NES-tail) intended as dominant-negative alleles that would compete with endogenous myosin VI for functional interactions in the respective subcellular compartments. Whereas expression of nuclear NLS-tail caused significant degradation of nascent DNA, expression of cytoplasmic NES-tail had no effect (Fig. 4f, Supplementary Fig. 5g), strongly suggesting that the compartment relevant for myosin VI activity in fork protection is the nucleus rather than the cytoplasm.

## Myosin VI promotes replication stress-induced WRNIP1 accumulation at replication forks

The requirement of myosin VI´s motor domain for its function in fork protection implied a mobility-dependent mechanism (Fig. 2e). This might involve an active transport of fork-protecting factors such as WRNIP1 toward stalled or reversed forks (Fig. 5a) or, alternatively, a transport of fork-destabilizing factors such as pertinent nucleases away from the sites of fork stalling (Fig. 5b). To differentiate between these models, we used SIRF to test whether myosin VI affected the recruitment of WRNIP1 to unperturbed or stalled replication forks. Consistent with our previous results (Fig. 3b), control cells expressing myosin VI afforded a WRNIP1 signal at unperturbed forks that increased after HU treatment (Fig. 5c). Knockdown of myosin VI did not significantly affect association of PCNA with replication forks (Fig. 5c, left panel) and WRNIP1 recruitment to unperturbed replication forks. However, under conditions of replication stress, we scored a clear defect in WRNIP1 accumulation at forks upon depletion of myosin VI, arguing for a model where myosin VI positively regulates WRNIP1´s enhanced association with stressed replication forks (Fig. 5a). Conversely, WRNIP1 depletion did not affect localization of myosin VI to replication forks (Supplementary Fig. 6). In summary, these data show the requirement of myosin VI for efficient WRNIP1 localization to stalled replication forks.

## Discussion

Our findings connect the actin-based motor protein myosin VI to a defined pathway of replication fork protection that maintains genome stability under conditions of replication stress. Using an unbiased mass spectrometry approach in combination with in situ localization studies, we found myosin VI to accumulate at stalled replication forks in response to nucleotide depletion, and functional assays have revealed a contribution to the WRNIP1-mediated protection of stressed forks from nucleolytic attack by DNA2. Based on our placement of myosin VI activity downstream of a series of factors known to mediate fork reversal, such as RAD51, SMARCAL1, HLTF and ZRANB3[3–5], we postulate that myosin VI specifically acts on reversed forks; however, our data do not exclude an alternative fork geometry generated by the above mentioned remodelers. The notion that the motor domain of myosin VI is required for its function suggests a role in shuttling; however, as myosin VI has also been shown to act in an anchoring fashion[40], we cannot exclude a model where myosin VI stabilizes the fork protection complex at the junction between parental and reversed strands in a static manner. In addition to the motor domain, we found the UBDs of myosin VI to be functionally important (Fig. 2e). The identification of another ubiquitin-binding protein, WRNIP1, as a direct interactor of myosin VI and the stimulating effect of ubiquitin chains on this interaction strongly suggest a regulatory role of ubiquitin signaling in this particular pathway of replication fork protection (Fig. 3d–f). Although ubiquitin is known as a signaling molecule in virtually all cellular processes, its contributions to the replication stress response are still poorly understood. In contrast, DNA binding by myosin VI does not appear to be important in this context, as the relevant mutant did not cause any fork destabilization.

Beyond the functional interaction of myosin VI and WRNIP1, our data support and expand recent evidence for nuclear functions of the actin cytoskeleton in genome maintenance. Although we did not directly address nuclear F-actin, the requirement of the myosin VI motor domain for fork protection (Figs. 2e, 4f) strongly suggests a mechanism based on the interaction of the myosin with nuclear actin filaments rather than invoking an actin-independent mechanism. However, WASp, a positive regulator of ARP2/3 dependent actin-polymerization, was recently shown to modulate RPA-regulated signaling upon genotoxic insult[41]. The authors convincingly demonstrate an actin-independent role of WASp as a chaperone-like factor for RPA´s ssDNA binding. Likewise, we cannot rule out additional, actin-independent functions of myosin VI.

Finally, while technical limitations have so far precluded firm evidence against an influence of the cytoplasmic actin cytoskeleton on genome maintenance, our newly designed tools in combination with classical dominant-negative approaches have provided clear evidence for the relevance of the nuclear pool of myosin VI for fork protection while excluding myosin VI-related cytoplasmic signaling events.

The formation of actin filaments inside the nucleus upon replication stress, detected by Lamm et al.[14], raises speculations about the relevance of the unique minus-end directionality of myosin VI and the possible orientation of actin filaments forming in the vicinity of

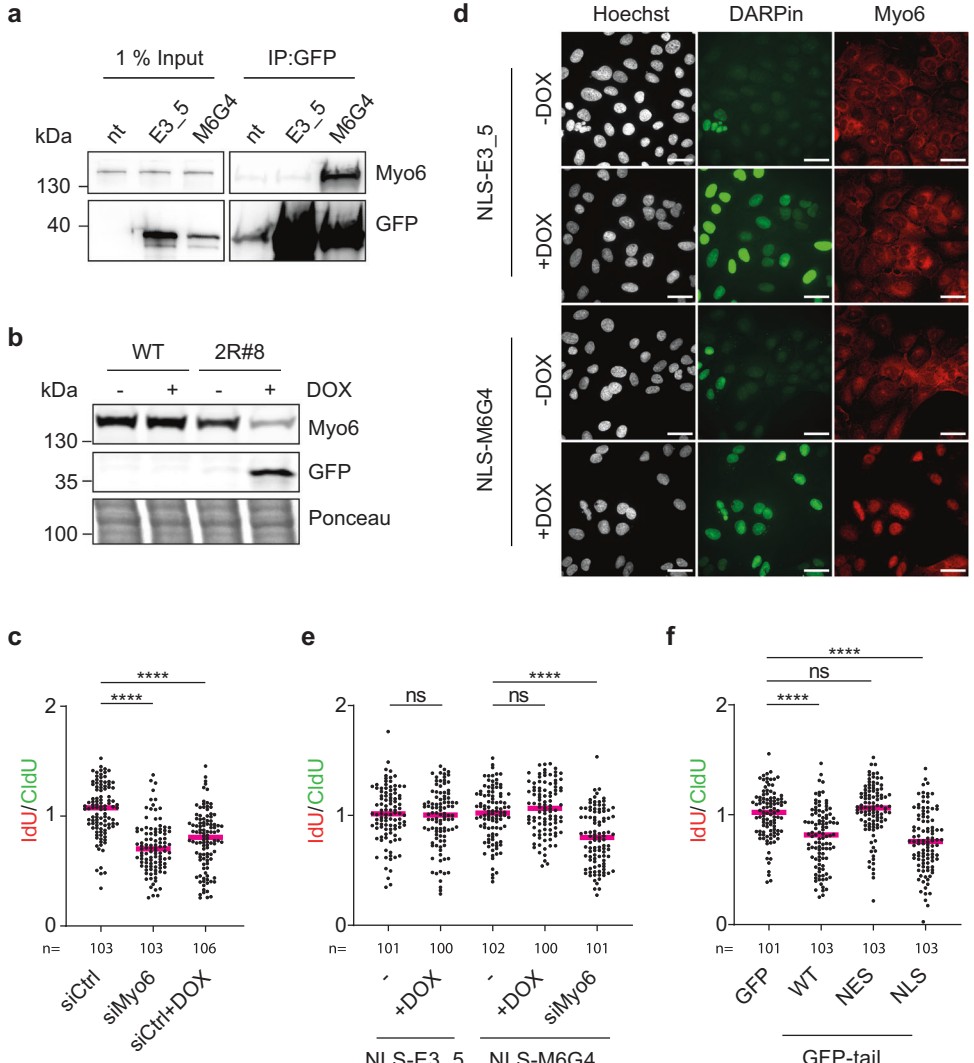

**Fig. 4 | Nuclear but not cytoplasmic myosin VI is active in fork protection.**
**a** DARPin M6G4 depletes myosin VI from cellular lysates. HEK293T cells, transfected with control (GFP-E3_5) or anti-myosin VI DARPin (GFP-M6G4), subjected to immunoprecipitations (IPs) against GFP, followed by western blotting. **b** DOX-induced degradation of myosin VI via a DARPin-based construct. A U2OS Flp-In T-REx single-cell clone harboring a DOX-inducible GFP-M6G4-2RING fusion construct (2R#8) was treated −/+ 20 ng/ml DOX for 24 h. Cellular lysates were analyzed by western blotting and Ponceau S staining. **c** DARPin-2RING fusion-mediated degradation of myosin VI interferes with fork protection. U2OS Flp-In T-REx cells harboring DOX-inducible GFP-M6G4-2RING (2R#8) were siRNA-transfected and treated −/+ 20 ng/ml DOX for 24 h, followed by fiber assays as shown in Fig. 2b. **d** DARPin-mediated re-localization of myosin VI to the nucleus. U2OS Flp-In T-REx cells harboring DOX-inducible GFP-M6G4-NLS or GFP-E3_5-NLS (control), treated −/+ 20 ng/ml DOX for 24 h, were analyzed by immunofluorescence (IF) using myosin VI-specific antibodies (red) and Hoechst (white). GFP-DARPins: green; scale bar = 40 μm. **e** Depletion of cytoplasmic myosin VI has no effect on fork stability. U2OS Flp-In T-REx cells harboring DOX-inducible GFP-M6G4-NLS or GFP-E3_5-NLS (control) were siRNA-transfected (siRNA#10 targeting myosin VI) and treated −/+ 20 ng/ml DOX for 24 h, followed by fiber assays as shown in Fig. 2b. **f** Inhibition of nuclear but not cytoplasmic myosin VI leads to fork de-stabilization upon replication stress. Fiber assays as shown in Fig. 2b, performed on U2OS cells transfected with compartment-specific GFP-tail constructs. For (**c**, **e**, **f**): IdU/CldU ratios are shown as dot plots with median values. Significance levels were calculated from the indicated number of fibers per sample using the two-tailed Mann–Whitney test (ns: not significant, ****: $p < 0.0001$). A representative experiment from three independent replicates is shown. Knockdown efficiencies and overexpression levels are shown in Supplementary Fig. 5. For (**a**, **b**, **d**): Results were confirmed by at least two independent experiments. Source data are provided as a Source Data file.

reversed forks. We also envision the involvement of other myosins, e.g., myosin I or myosin V[10], in fork protection, opening the possibility for a competition between minus- and plus-end-directed motors. Probing the role of other myosins as well as actin cytoskeleton proteins such as bundling, capping, assembly or disassembly factors will thus be important for future studies. Taken together, our discovery of the requirement of myosin VI-dependent transport or tethering for the protection of stressed replication forks, possibly controlled by ubiquitin binding, adds to the accumulating evidence for a key role of the nuclear actin cytoskeleton in genome maintenance and paves the way for exploring new layers of regulation of nuclear transactions by a set of proteins better known for their role in cytoplasmic signaling.

# Methods

## Cell lines, cultivation and treatments

U2OS, HeLa and HEK293T cells were maintained in DMEM containing 10% fetal bovine serum, L-glutamine (2 mM), penicillin (100 U/ml), and streptomycin (100 μg/ml) (Thermo Fisher Scientific). U2OS Flp-In T-REx cell lines were maintained in DMEM containing 10% fetal bovine serum, L-glutamine, penicillin, streptomycin and blasticidin (5 μg/ml) (Invivogen). All cell lines were cultured in humidified incubators at 37 °C with 5% $CO_2$. Treatments were performed with hydroxyurea (4 mM, Merck), the MRE11 inhibitor mirin (25 μM, Merck) or the DNA2-specific inhibitor C5 (25 μM, AOBIOUS) for 5 h. For SILAC labeling, HeLa cells were cultured for at least 5 passages in SILAC DMEM (Invitrogen)

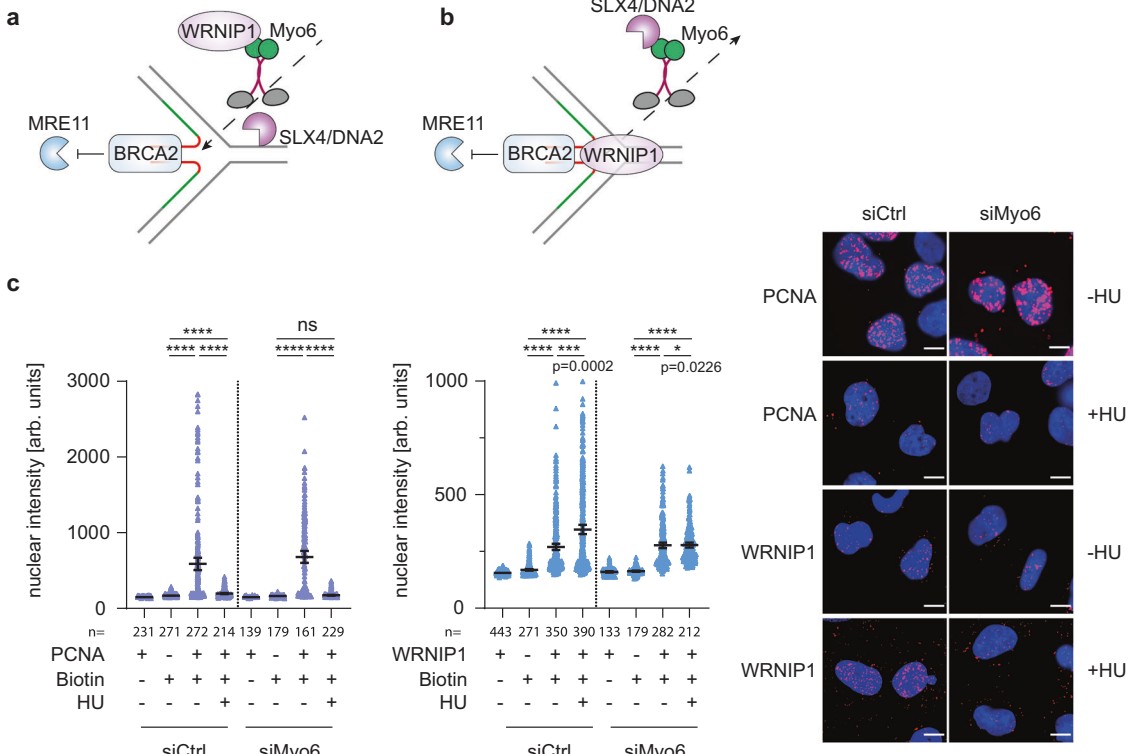

**Fig. 5 | Myosin VI is required for efficient localization of WRNIP1 to stalled forks. a, b** Models of how myosin VI could mediate fork protection in its role as a motor protein. **c** Myosin VI depletion interferes with efficient fork localization of WRNIP1. U2OS cells were siRNA-transfected, followed by SIRF assays as indicated. Left: dot plots of PLA signal intensities with mean values −/+ 95% confidence intervals. Significance levels were calculated using the two-tailed Mann–Whitney test from indicated number of nuclei per sample (ns, non-significant, ****: $p < 0.0001$, ***: $p < 0.001$, *: $p < 0.05$) and annotated for $p > 0.0001$. Right: representative images, Hoechst (blue), PLA (magenta), scale bar = 10 μm. A representative experiment from three independent replicates is shown. Knockdown efficiency is shown in Supplementary Fig. 6. Source data are provided as a Source Data file.

supplemented with dialyzed FBS (Invitrogen) and containing either L-arginine and L-lysine (Merck) or L-arginine [13C6] and L-lysine [2H4] (Cambridge Isotope Laboratories).

### Transfections

For overexpression purposes, HEK293T were transfected using polyethyleneimine (PEI) (Polysciences). Other cell types were transfected using Fugene HD (Promega) or Lipofectamine 2000 (Life technologies) according to the manufacturer´s instructions. All expression constructs used in this study are listed in Supplementary Table 1.

For knockdowns, cells were transfected with siRNAs using Lipofectamine RNAiMAX (Life Technologies) according to the manufacturer´s instructions at a final RNA concentration of 20 nM for 72 h. Knockdown of myosin VI was achieved with a pool of 4 different siRNAs (Hs_MYO6_5 FlexiTube siRNA, Hs_MYO6_7 FlexiTube siRNA, Hs_MYO6_8 FlexiTube siRNA and Hs_MYO6_10 FlexiTube, Qiagen). For rescue experiments in the U2OS Flp-In cell line expressing GFP-myosin VI, a single siRNA targeting the 3´-UTR of the myosin VI transcript (Hs_MYO6_10 FlexiTube siRNA) was used. RAD51 and ZRANB3 knockdowns were performed using a pool of two independent siRNAs each. A list of all siRNAs used in this study can be found in Supplementary Table 3.

### Generation of stable cell lines

U2OS Flp-In T-REx cell lines for DOX-inducible expression were generated by co-transfection of the respective pDEST-FRT-TO construct with the pOG44 Flp-Recombinase (Supplementary Table 1). 24 h post-transfection, cells were selected with 100 μg/ml hygromycin (Invivogen) for 10 days. Hygromycin-resistant cells were sorted for GFP-positive clones using a BD FACS Aria III SORP instrument. Single-cell

clones were tested for construct expression and myosin VI depletion after DOX treatment by western blotting using GFP- and myosin VI-specific antibodies (Supplementary Table 2).

### Generation of plasmids

Fragments were inserted via restriction/ligation cloning or following PCR amplification with specific oligonucleotides, listed in Supplementary Table 4. For Gateway cloning, Gateway® LR Clonase® II enzyme mix (Thermo Fisher Scientific) was used according to the manufacturer´s instructions. Detailed information about individual constructs will be provided upon request.

### Site-directed mutagenesis

Site-directed mutagenesis was performed using Pfu Turbo DNA Polymerase (Agilent). The amplification product was digested with DpnI (New England Biolabs), *E. coli* TOP10 cells were transformed with the construct followed by sequence verification. Oligonucleotides for mutagenesis are listed in Supplementary Table 4.

### Protein production and purification

GST fusion proteins were produced in *E. coli* Bl21 (DE3) cells at 37 °C for 4 h after induction with 1 mM IPTG (Generon) at an $OD_{600}$ of 0.8. Cells were pelleted and lysed by sonication in PBS/0.1% Triton X-100 (Merck) supplemented with protease inhibitor cocktail (SIGMAFAST). Clarified supernatants were incubated with 1 ml of GSH-Sepharose beads (Cytiva) per liter of bacterial culture. After 2 h at 4 °C, the beads were washed with PBS/0.1% Triton X-100 and maintained in storage buffer (50 mM Tris, pH 7.4, 100 mM NaCl, 1 mM EDTA, 1 mM DTT, and 10% glycerol).

Expression of DARPins with N-terminal MRGS(H)$_8$ tag, USP2cc with N-terminal MRGS(H)$_8$ tag, myosin VI (aa 992-1122) with N-terminal

MRGS(H)$_8$ and C-terminal Avi tag in *E. coli* BL21 (DE3) was induced with 1 mM IPTG for 20 h at 18 °C. Cells were resuspended in buffer A (50 mM Tris-HCl pH 7.4, 250 mM NaCl, 10% glycerol, 1 mM DTT, 20 mM imidazole) and lysed by sonication. The clarified supernatant was subjected to affinity chromatography on Ni-NTA resin (Qiagen), and eluted protein was rebuffered using PD 10 columns (Cytiva) in storage buffer (50 mM Tris, pH 7.4, 100 mM NaCl, 1 mM EDTA, 1 mM DTT, and 10% glycerol).

Myosin VI (aa 992-1122) with N-terminal MRGS(H)$_8$ and C-terminal Avi tag was biotinylated in vivo by co-expressing biotin-ligase BirA (pBirAcm from Avidity) in *E. coli* BL21 (DE3). 50 µM biotin was added to the growth medium (LB) before induction with IPTG.

## GST-pulldown assay coupled to mass spectrometry

For SILAC experiments, $8 \times 10^7$ HeLa cells were lysed in 2 ml JS buffer (100 mM HEPES pH 7.5, 50 mM NaCl, 5% glycerol, 1% Triton X-100, 2 mM MgCl$_2$, 5 mM EGTA, 1 mM DTT), supplemented with protease inhibitor cocktail (SIGMAFAST) and Benzonase® (Merck). 50 µg of GST and 70 µg of GST-MyUb fusion protein immobilized on 50 µl GSH-Sepharose beads were incubated with 1 ml of cellular lysate for 2 h at 4 °C. Beads were washed five times in 1 ml JS buffer. Labels were switched in 2 out of 4 biological replicates. SILAC samples were pooled during the last wash. Bound proteins were eluted in 2× NuPAGE LDS Sample Buffer (Life Technologies) supplemented with 1 mM dithiothreitol, heated at 70 °C for 10 min, alkylated by addition of 5.5 mM chloroacetamide for 30 min, and separated by SDS-PAGE on a 4–12% gradient Bis–Tris gel (Invitrogen). Proteins were stained using the Colloidal Blue Staining Kit (Life Technologies) and digested in-gel using 0.6 µg of MS-approved trypsin (Serva) per gel fraction. Peptides were extracted from the gel and desalted using reversed-phase C18 StageTips.

Peptide fractions were analyzed on a quadrupole Orbitrap mass spectrometer (Q Exactive Plus, Thermo Fisher Scientific) equipped with a UHPLC system (EASY-nLC 1000, Thermo Fisher Scientific). Peptide samples were loaded onto C18 reversed-phase columns (25 cm length, 75 µm inner diameter, 1.9 µm bead size, packed in-house) and eluted with a linear gradient from 1.6 to 52% acetonitrile containing 0.1% formic acid in 90 min. The mass spectrometer was operated in a data-dependent mode, automatically switching between MS and MS2 acquisition. Survey full scan MS spectra (m/z 300–1,650, resolution: 70,000, target value: 3e6, maximum injection time: 20 ms) were acquired in the Orbitrap. The 10 most intense ions were sequentially isolated, fragmented by higher energy C-trap dissociation (HCD) and scanned in the Orbitrap mass analyzer (resolution: 35,000, target value: 1e5, maximum injection time: 120 ms, isolation window: 2.6 m/z). Precursor ions with unassigned charge states, as well as with charge states of +1 or higher than +7, were excluded from fragmentation. Precursor ions already selected for fragmentation were dynamically excluded for 20 s.

Raw data files were analyzed using MaxQuant (version 1.5.2.8)[42]. Parent ion and MS2 spectra were searched against a reference proteome database containing human protein sequences obtained from UniProtKB (HUMAN_2016_05) using the Andromeda search engine[43]. Spectra were searched with a mass tolerance of 4.5 ppm in MS mode, 20 ppm in HCD MS2 mode, strict trypsin specificity, and allowing up to two mis-cleavages. Cysteine carbamidomethylation was searched as a fixed modification, whereas protein N-terminal acetylation, methionine oxidation, GlyGly (K), and N-ethylmaleimide modification of cysteines (mass difference to cysteine carbamidomethylation) were searched as variable modifications. The Re-quantify option was turned on. The dataset was filtered based on posterior error probability (PEP) to arrive at a false discovery rate of below 1%, estimated using a target-decoy approach[44]. Statistical analysis and MS data visualization were performed using the R software environment (version 4.2.1). Potential contaminants, reverse hits, hits only identified by site and hits with no unique peptides were excluded from the analysis. Statistical significance was calculated using a moderated *t*-test (limma package)[45].

GO term analysis (biological process) was performed using EnrichR[46]. Visualized GO terms were selected based on adjusted *p* value, odds ratio and semantic uniqueness. To determine the number of nuclear proteins among MyUb interactors (fold change > 2, FDR < 0.05), GO cellular component annotations were retrieved from the STRING network tool[47].

## Preparation of unanchored K63-linked polyubiquitin chains

Unanchored K63-linked polyubiquitin chains were prepared by incubating 0.05 µM E1 ($^{His}$Uba1), 2 µM $^{His}$Ubc13-Mms2 and 0.5 µM E3 (Pib1RING+100aa)[48] in a 1 ml reaction containing 40 mM HEPES, pH 7.4, 8 mM magnesium acetate, 50 mM NaCl and 30 µM ATP. *Wildtype* ubiquitin (purified bovine ubiquitin, Sigma) was used at a concentration of 8 µM and 4 µM of ubiquitin mutant K63R was added for capping of the chains. The reaction was incubated for 1.5 h at 30 °C and 1–4 µl of the chain reaction were used in GST-pulldown assays.

## GST-pulldown assays

GST-pulldown assays were performed with lysates from $5 \times 10^6$ unlabeled HeLa cells, and interactors were detected by western blotting using antibodies against endogenous proteins.

To identify DARPins suitable for pulldown assays, screening was performed by incubating 10 µg GST (as control) or 14 µg GST-MyUb immobilized on 20 µl GSH-Sepharose beads with a final DARPin concentration of 1 µM in 200 µl PBS/0.1% Triton X-100. Beads were washed three times in 1 ml PBS/0.1% Triton X-100, boiled for 10 min in NuPAGE® LDS Sample Buffer and subjected to SDS-PAGE. Detection was performed using Instant Blue protein stain (Biozol).

To identify direct protein-protein interactions, we performed pulldown assays by incubating 5 µg GST or GST-WRNIP1 immobilized on 20 µl GSH-Sepharose beads with various concentrations of His-MIUMyub domain in 200 µl modified JS buffer (100 mM HEPES pH 7.5, 50 mM NaCl, 5% glycerol, 1% Triton X-100, 2 mM MgCl$_2$, 5 mM EGTA, 1 mM DTT). Beads were washed three times in 1 ml modified JS-buffer, boiled for 10 min in NuPAGE® LDS Sample Buffer and subjected to SDS-PAGE and subjected to western blotting.

## Immunoprecipitation of GFP-tagged proteins

HEK293T cells were PEI-transfected with the respective plasmid (Supplementary Table 1) for 24 h, followed by lysis in JS buffer (100 mM HEPES pH 7.5, 50 mM NaCl, 5% glycerol, 1% Triton X-100, 2 mM MgCl$_2$, 5 mM EGTA, 1 mM DTT) supplemented with protease inhibitor cocktail (SIGMAFAST) and Benzonase®. Cell lysates were cleared by centrifugation for 30 min at 4 °C and incubated with GFP-trap magnetic agarose beads (Chromotek) for 1 h at 4 °C. After 3 washes with JS buffer, beads were boiled for 10 min in NuPAGE® LDS Sample Buffer and subjected to western blotting.

## Proteasome inhibition

U2OS Flp-In T-REx cells harboring DOX-inducible GFP-M6G4-2RING (2R#8) were treated with 5 µM MG-132 (Enzo Life Sciences) for 24 h in the presence of 2 µg/ml DOX.

## iPOND

U2OS cells were labeled with 10 µM EdU (Merck) for 30 min. Subsequently, cells were fixed with 1% formaldehyde (Merck) for 10 min, followed by quenching with 125 mM glycine (Merck) for 10 min. After two washing steps with PBS/1% BSA, cells were collected by scraping, followed by permeabilization in PBS/0.1% Triton X-100. Subsequently, cells were washed with PBS/1% BSA and subjected to the Click-iT reaction in a solution containing 10 mM sodium ascorbate (Merck), 0.1 mM azide-PEG$_3$-biotin conjugate (Merck) and 2 mM copper sulfate (Merck) for 30 min at room temperature. Cells were then washed twice in PBS/1% BSA, lysed in 10 mM Tris-HCl pH 8.0, 140 mM NaCl, 1% Triton X-100, 0.1% sodium deoxycholate, 0.1% SDS, supplemented with

SIGMAFAST protease inhibitor cocktail, and sonicated using a Bioruptor (Diagenode). Lysates were cleared by centrifugation for 45 min at 4 °C in a table-top centrifuge and subjected to streptavidin-agarose beads (Thermo Fisher Scientific) overnight at 4 °C. The next day, beads were washed five times in PBS/1% BSA and de-crosslinking was carried out for 30 min in NuPAGE® LDS Sample Buffer at 95 °C. For protein detection, samples were subjected to SDS-PAGE and immunoblotted with relevant antibodies (Supplementary Table 2).

## Immunofluorescence

For immunofluorescence analysis, cells were fixed with 4% paraformaldehyde (Merck) for 10 min, permeabilized for 5 min at room temperature with 0.1% Triton X-100 and incubated for 1 h in PBS/3% BSA. Subsequently, cells were incubated with primary antibodies for 1 h (α-myosin VI α-rabbit in a 1:400 dilution), followed by 3 × 5 min washing steps with PBS/0.1% Triton X-100 and incubation with secondary antibodies for 30 min at room temperature. Coverslips were mounted with ProLong™ Diamond Antifade Mountant (Thermo Fisher Scientific). Images were acquired using the Leica Application Suite X version 3.7.5.24914 on a Leica AF-7000 widefield microscope and analyzed with ImageJ 153t.

## Confocal microscopy

For confocal microscopy, U2OS cells were seeded in μ-Slide 8 Well Chamber Slides (Ibidi) with a confluency of 80% ($5 \times 10^4$ cells per well). To visualize actin filaments, cells were fixed in 4% paraformaldehyde for 10 min at room temperature and permeabilized using 0.3% Triton-X for 10 min. For F-actin stainings, cells were incubated with Alexa Fluor 647 Phalloidin (1:100) (Fisher Scientific) and Hoechst (1:10.000) (Merck) in PBS for 1 h, followed by three washing steps of 5 min each with PBS. Samples were imaged using the Fusion 1.1.0.1 software on a BC43 Spinning Disk Confocal (Oxford Instruments) microscope using blue (405 nm), green (488 nm) and red (612 nm) excitation wavelengths. A 60× oil objective lens was chosen. Z-stack imaging was performed with 30–40 steps in 0.3 μm (Phalloidin) or 0.4 μm increments (GFP-myosin VI) and a z-plane between #8 and #17 was chosen for nuclear actin quantification using Fiji ImageJ 153t software.

## Immunoblotting

Samples were separated via SDS-PAGE and transferred to nitrocellulose membranes using the Trans-Blot Turbo® system (Bio Rad). Membranes were blocked for 1 h at room temperature in 5% milk/PBS/0.1% TWEEN-20 and incubated with primary antibodies in a 1:1000 dilution in PBS/0.1% TWEEN-20/1% BSA; either for 1 h at room temperature or overnight at 4 °C. Afterwards, membranes were washed with PBS/0.1% TWEEN-20 and incubated with secondary antibodies for 1 h at room temperature. Detection was performed by enhanced chemiluminescence using a Fusion FX (Vilber Lourmat) instrument with the Fusion Capt Advance Fx7 17.03 software after incubation with HRP-coupled secondary antibodies or by direct fluorescence using an Odyssey Clx imaging system (LI-COR) with the Image Studio version 3.1 software after incubation with secondary antibodies coupled to a fluorescent dye (Supplementary Table 2). Uncropped images from the main figures can be found in the 'Source Data' file provided with this paper.

## Fiber assays

U2OS cells were labeled with 50 μM CldU (Merck) for 20 min and 50 μM IdU (Merck) for 20 min, respectively. Cells were trypsinized, resuspended in PBS and diluted to $1.75 \times 10^5$ cells/ml. Labeled cells were mixed with unlabeled cells at a ratio of 1:1. Lysis of the cells was carried out directly on microscopy slides, where 4 μl of the cells was mixed with 7.5 μl of lysis buffer (200 mM Tris-HCl pH 7.4, 50 mM EDTA, 0.5% SDS). After 9 min, the slides were tilted at an angle of 15–45° and the DNA fibers were stretched on the slides. The fibers were fixed in methanol/acetic acid (3:1) overnight at 4 °C. Following fixation,

the DNA fibers were denatured in 2.5 M HCl for 1 h, washed with PBS and blocked with PBS/0.1% TWEEN-20/2% BSA for 40 min. The fibers were incubated with primary antibodies against CldU (Rat monoclonal anti-BrdU (clone BU1/75 (ICR1), Abcam) and IdU (Mouse monoclonal anti-BrdU (clone B44), BD Biosciences) (1:50 dilution) for 2.5 h, washed with PBS/0.1% TWEEN-20 and incubated with secondary antibodies labeled with Alexa Fluor 488 and Alexa Fluor 647 (1:100 dilution). The slides were mounted in ProLong™ Diamond Antifade Mountant. Images of the DNA fibers were acquired using a Leica Thunder widefield microscope and analysis was carried out using Fiji ImageJ 153t. To assess overall replication speed, only fibers where both tracks had equal length were measured. To assess fork asymmetry (IdU/CldU ratio), also fibers with shorter IdU tracks were analyzed and analyzed using GraphPad Prism 8.4.0 (538).

## Proximity ligation assays (PLA)

U2OS cells were seeded on coverslips with a confluency of 80%. Afterwards, cells were fixed in 4% paraformaldehyde for 10 min and permeabilized in 0.3% Triton X-100 for 10 min. PLA was then carried out using the Duolink® In Situ Red starter kit (Merck) according to the manufacturer's instructions. Primary antibodies were used in a 1:100 dilution (α-WRNIP1 α-rabbit, α-Myosin VI α-mouse). In addition, Hoechst staining was included prior to mounting coverslips in Pro-Long™ Diamond Antifade Mountant. Images were acquired using a Leica Thunder widefield microscope and analysis was carried out using Fiji ImageJ 153t.

## In situ analysis of protein interactions at DNA replication forks (SIRF)

For SIRF, cells were pulsed with 10 μM EdU for 10 min and then left untreated or treated with 4 mM HU for 5 h. After fixation in 4% paraformaldehyde for 10 min and permeabilization in 0.3% Triton X-100 for 10 min, the Click-iT reaction was performed for 1 h at room temperature in PBS containing 2 mM copper sulfate, 10 μM azide-PEG$_3$-biotin conjugate and 100 mM sodium ascorbate. PLA was then carried out as described above. Primary antibodies were used in a 1:100 (α-WRNIP1 α-rabbit, α-Myosin VI α-mouse, α-Biotin α-mouse) or 1:1000 dilution (α-PCNA, α-rabbit).

## DARPin selection and initial screening

To generate myosin VI-specific DARPins, biotinylated myosin VI (aa 992-1122) isoform 1 with N-terminal MRGS(H)$_8$ and C-terminal Avi tag ($^{His}$myosin VI (aa 992-1122)$^{Avi}$) was immobilized on either MyOne T1 streptavidin-coated beads (Pierce) or Sera-Mag neutravidin-coated beads (GE Healthcare). The use of the type of beads was alternated during selection rounds. Ribosome display selections were performed essentially as described[49], using a semi-automatic KingFisher Flex MTP 96-well platform. Although DARPin-screening was performed to isolate isoform 1-specific binders, DARPin (M6G4), which showed a biological effect, was characterized as pan-isoform-specific (Supplementary Fig. 4c).

The library includes N3C DARPins, consisting of three internal and randomized ankyrin repeats as described earlier[38]. The originally described C-cap was replaced with a C-cap showing better stability toward unfolding implementing mutations in 5 amino acid positions[36,50,51] to facilitate downstream experiments like protein fusions. Additionally, we introduced a second randomization strategy in the N- and C-cap as described[36,52] to also allow interaction of the capping repeats with the target. The libraries of DARPins with randomized and non-randomized N- and C- terminal caps, both containing randomized internal repeats and a stabilized C-cap, were mixed in a 1:1 stoichiometry to increase diversity. Successively enriched DARPin pools were cloned as intermediates in a ribosome display vector[52]. Selections were performed over four rounds with decreasing target concentration and increasing washing steps to enrich for binders with

slow off-rates and thus high affinities. The first round accomplished the initial selection against myosin VI at low stringency. The second round included pre-panning with the undesired myosin VI isoforms 2 (aa 992-1099) and 3 (aa 992-1131) immobilized on magnetic beads, with the supernatant transferred to the immobilized desired target myosin VI isoform 1. The third round included this pre-panning and the addition of non-biotinylated myosin VI isoform 1 to enrich for binders with slow off-rates. The fourth and final round included the pre-panning step and selection was performed with low stringency to collect all binders.

The final enriched pool was cloned as fusion construct with an N-terminal MRGS(H)$_8$ tag and C-terminal FLAG tag via unique BamHI and HindIII sites into a bacterial pQE30 derivative vector containing lacI$^q$ for expression control. After transformation of *E. coli* XL1-blue, 380 single DARPin clones were expressed in 96-well format and cells were lysed by addition of B-Per Direct detergent plus lysozyme and nuclease (Pierce). The resulting bacterial crude extracts of single DAR-Pin clones were subsequently used in a Homogeneous Time Resolved Fluorescence (HTRF)-based screen to identify potential binders. The clone M6G4 that was selected for downstream applications was monoclonalized, by cutting the DARPin ORF, re-ligating it in fresh vector, retransformation and sequence verification. Binding of the FLAG-tagged DARPins to streptavidin-immobilized biotinylated $^{His-Avi}$myosin VI (aa 992-1122) was measured using FRET (donor: Streptavidin-Tb cryptate (610SATLB, Cisbio), acceptor: mAb anti FLAG M2-d2 (61FG2DLB, Cisbio)). Further HTRF measurement against 'No Target' allowed for discrimination of myosin VI isoform 1-specific hits. Experiments were performed at room temperature in white 384-well Optiplate plates (PerkinElmer) using the Taglite assay buffer (Cisbio) at a final volume of 20 µl per well. FRET signals were recorded after an incubation time of 30 min using a Varioskan LUX Multimode Microplate (Thermo Fisher Scientific). HTRF ratios were obtained by dividing the acceptor signal (665 nm) by the donor signal (620 nm) and multiplying this value by 10,000 to derive the 665/620 ratio. The background signal was determined by using reagents in the absence of DARPins.

### Surface plasmon resonance

Surface plasmon resonance (SPR) experiments were performed on a Biacore X100 system, equilibrated at 25 °C in HBS-EP buffer (10 mM HEPES pH 7.4, 150 mM NaCl, 3 mM EDTA, 0.005% v/v Surfactant P20, Cytiva) using a streptavidin-coated sensor chip (CAP) and biotinylated MIUMyUb-domain as immobilized target with a density of 60-80 RU. The Biacore X100 Control Software version 2.0.2 was used for data acquisition and the Biacore X100 Evaluation version 2.0.2 for data analysis. DARPins were injected for 180 s at a flow rate of 30 µl/min in increasing concentrations (factor 1.5) ranging from 6.5 nM to 0.25 µM. Kinetic data ($K_D$, $k_{on}$ and $k_{off}$) for the M6G4 DARPin were obtained using the fitting tool (1:1 binding model) of the Biacore X100 evaluation software version 2.0.2 and are reported as the mean of four independent experiments with corresponding standard deviations. The control DARPin (E3_5) did not show any binding to the biotinylated MIUMyUb-domain in our measurements.

### Reporting summary

Further information on research design is available in the Nature Portfolio Reporting Summary linked to this article.

## Data availability

All reagents used in the paper are listed in Supplementary Table 5. Parent ion and MS2 spectra were searched against a reference proteome database containing human protein sequences obtained from UniProtKB (HUMAN_2016_05) using the Andromeda search engine. The mass spectrometry-based proteomics data have been deposited to the ProteomeXchange consortium via the PRIDE partner repository[53] with the data set identifier PXD035394. Source data are provided with this paper.

## Code availability

Custom codes used for the preparation of volcano and GO term plots as well as the Image J-based quantifications are available on GitHub [https://github.com/helle-ulrich-lab/myosinVI-replication-fork-stability].

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

## Acknowledgements

We thank Ronald Wong and Kirill Petriukov for scientific discussions and Katharina Schlacher, Lorenza Penengo and Massimo Lopes for help with setting up the fiber assay. We thank Ron Hay, Simona Polo, Lorenza Penengo and Christian Renz for sharing constructs, cell lines or recombinant proteins. We thank Thomas Reinberg, Sven Furler and Joana Marinho from HT-BSF UZH for their assistance in performing the ribosome display DARPin selection and screening. The IMB Core Facilities for Proteomics, Flow Cytometry, Microscopy and Protein Production are acknowledged for technical support and reagents. This work was funded by the Deutsche Forschungsgemeinschaft (DFG, German Research Foundation)—Project-ID 393547839—SFB 1361 awarded to H.D.U. and P.B., Project-ID BE 5342/2-1—FOR 2800 awarded to P.B. and Project-ID 408799149 awarded to H.P.W.

## Author contributions

J.Sh., H.D.U., and H.P.W. conceived the study, J.Sh., V.A., K.H., S.M., T.S., and H.P.W. performed experiments and data analysis in cell and molecular biology, I.M., J.B.H., and P.B. performed mass spectrometry experiments and data analysis, J.Sc., B.D., and A.P. designed and supervised the DARPin selections, H.D.U. and H.P.W. wrote the paper and created the figures, and all authors discussed the data and provided input during paper preparation.

## Competing interests

The authors declare no competing interests.
