## [Peer Review File · Nature Communications]

Nuclear myosin VI maintains replication fork stabilityREVIEWER COMMENTS

Reviewer #1 (Remarks to the Author):

The manuscript by Shi et al describes the contribution of nuclear myosin VI to replication fork stability. The authors have performed a significant amount of work and directly addressed that nuclear myosin VI supports fork stability. This quality of work is highly important to take the nuclear myosin (and actin) field forward. As is typical with this field, the authors have had to perform extensive controls to show there is a nuclear myosin role rather than actually determining the role of the myosin.

There is no doubt that this work is at a high standard and the manuscript is very well-written. Moreover, I enjoyed reading the paper and wish to see it published. However, my concerns arise from that fact that besides concluding that myosin VI is involved in fork stability, there is very little conclusion as to what function it may have. Therefore, there is a lack of mechanistic insight, despite a very large body of work. This is not a criticism of the work presented or the conclusions drawn from the body of work. Importantly, this work does create a strong platform for future work.

I have some specific comments:

- (a) You mention actin several times in the manuscript. What happens to fork stability (your assays) when you perturb nuclear actin?
- (b) What happens to the actin and myosin VI nuclear levels during replication stress?
- (c) Can you express the recombinant WRNIP to determine if there is a direct MVI interaction?
- (d) Is the interaction with WRNIP mediated via Ub? Can you manipulate this process to conclude how the proteins interact?
- (e) You strongly hint towards a motility-based role of myosin VI. I would tone down some of these statements because there is not experimental data to support this conclusion.

Reviewer #2 (Remarks to the Author):

I was specifically asked to look into the mass spectrometry aspects of the paper. The methodology is sound. SILAC was used and 4 GST pull downs repeats were performed including reverse labelling. The methods section is thorough. There are nonetheless a few minor elements to provide in the methods:
-please specify if dialysed serum was used in SILAC labeling of HeLas (line 234)
-please specify the trypsin concentration used (line 297)
-please specify how the 4 repeats were compared and which hits were kept in the final list, i.e. specify which hits were common to all repeats or if less, how many?

Reviewer #3 (Remarks to the Author):

The authors have studied the role of myosin VI in chromosome replication. They provide evidence that i) Myosin VI interacts with replisome components ; ii) Myosin VI depletion affects fork elongation and results into fork processing ; iii) Myosin VI cooperates with WRNIP1 at forks.

General comments

In general, the finding that nuclear Myosin VI influences chromosome replication is original and potentially relevant. Most of the results are convincing. However, there are serious issues that must be addressed.

Specific comments

1. The field of reversed forks is already confusing and controversial. The authors abuse of the term reversed fork without providing any direct evidence that forks indeed undergo reversal. The authors have two options: a) if they really want to push the fork reversal concept, then they need to analyze the replication intermediates by EM. b) Remove any statement related to fork reversal from abstract and results sections (including the schemes in the figures). They can eventually speculate about fork reversal in the discussion without being too dogmatic in their conclusions.

2. It would be important to test whether the impact on replication caused by Myosin VI depletion is due to an in trans effect resulting from ATR activation. In other words, does Myosin VI depletion activate ATR? And if yes, does ATR activation play any role in fork stalling/slowing down following Myosin VI depletion?

3. Figure 4: I do not find the result convincing. This part could be removed.

Marco Foiani

Reviewer #4 (Remarks to the Author):

In this manuscript, the authors show the role of nuclear myosin VI in the stabilisation of replication forks. To do that, they identified nuclear interactors of myosin VI by mass spectrometry analysis. They also developed DARPin-specific binders to manipulate the localisation of myosin VI and to confirm the involvement of nuclear myosin VI in controlling the fork stability. This is an original study using different methods to show what is the role of myosin VI on replication forks' stability.

As requested, I particularly focused on the experiment involving DARPins screening and selection to obtain the myosin VI binders.

1) Concerning the selection of DARPins by ribosome display, the authors use "undesired myosin VI isoforms 2 and 3" as competitors, but no data is shown to actually demonstrate the specificity of the selected DARPins towards isoform 1. The authors should provide data to show the isoform 1 specificity of M6G4.

2) The authors should clearly indicate in the selection protocol by ribosome display what are the constructs employed for myosin VI isoforms 2 and 3 (full length proteins or truncated proteins? which amino acids were used?).

3) Throughout the entire manuscript, the authors mention myosin VI, however there are different isoforms and the DARPins are theoretically isoform 1 specific. Therefore, for clarity purposes, the authors should indicate in the introduction the differences between the three isoforms (localisation, expression, function?) and explain why they selected DARPins against the isoform 1 only.

4) While the strategy used by the authors to screen their binders after ribosome display is interesting (HTRF), no data is available to show this. Therefore, the authors should add the data of the HTRF screening in a figure and explain these results within the results section (I139: "we obtained 54 candidates from the library...").

5) The authors should determine the Kd of their lead DARPin (M6G4) to confirm the panning strategy used by the authors "I430: increasing washing steps to enrich for binders with slow off-rates and thus high affinities".

6) The authors should indicate in a supplementary file, the DNA and protein sequences of the anti-myosin VI DARPin M6G4.

7) The authors should show whether the binding of DARPin M6G4 affects myosin VI PPIs such as WRNIP1 (if any inhibition, it would probably be by steric hindrance rather than direct PPI inhibition). This is an important question to answer before using the DARPin M6G4 as relocalisation tool of myosin VI.

Comments to the authors unrelated to the DARPin screening and selection:

8) Abstract: l27-28: "Using nuclear localization sequence (NLS) and ubiquitin E3-fusion DARPins to manipulate myosin VI levels in a compartment-specific manner". This sentence sounds like the targeted degradation of myosin VI by DARPin-E3 ligase fusion is compartment-specific, which is not the case here. This sentence should be rephrased.

9) Fig3b: a proteasome inhibitor control would be required to show that the degradation of myosin VI by the DARPin-RINGx2 construct is proteasome dependent.

10) L158-161: the authors should directly test the NES-DARPin M6G4 construct to exclude the endogenous myosin VI from the nucleus. Even if this experiment does not work (with the NES-DARPin M6G4), it is important to show it rather than a test case with a GFP and an anti-GFP DARPin. The argument stating that the NES-DARPin fusion is synthesised in the cytosol is correct but the GFP (or myosin VI) are also synthesised in the cytosol, so the DARPins should be able to capture the GFP/myosin VI before it goes into the nucleus.

Response to the reviewers' comments

We would like to thank the referees for carefully evaluating our work and providing constructive comments, which helped us to significantly improve the quality of our manuscript. We are confident that our revisions successfully address their previous concerns.

Please find a point-by point response below.

Reviewer #1

The manuscript by Shi et al describes the contribution of nuclear myosin VI to replication fork stability. The authors have performed a significant amount of work and directly addressed that nuclear myosin VI supports fork stability. This quality of work is highly important to take the nuclear myosin (and actin) field forward. As is typical with this field, the authors have had to perform extensive controls to show there is a nuclear myosin role rather than actually determining the role of the myosin.

There is no doubt that this work is at a high standard and the manuscript is very well-written. Moreover, I enjoyed reading the paper and wish to see it published. However, my concerns arise from that fact that besides concluding that myosin VI is involved in fork stability, there is very little conclusion as to what function it may have. Therefore, there is a lack of mechanistic insight, despite a very large body of work. This is not a criticism of the work presented or the conclusions drawn from the body of work. Importantly, this work does create a strong platform for future work.

We thank this reviewer for this very positive assessment of our work. By expanding on the physical interactions between myosin VI, WRNIP1 and ubiquitin (see below), we have now added further mechanistic details in support of the model that we put forth in our discussion.

I have some specific comments:

(a) You mention actin several times in the manuscript. What happens to fork stability (your assays) when you perturb nuclear actin?

The reviewer's question is likely based on the expectation that perturbation of actin should result in a phenotype similar to myosin VI inactivation if the two factors indeed cooperate. Based on this model, we have tested the consequences of interfering with actin in two different fibre assay setups:

(I) To interfere specifically with **nuclear** actin polymerization, we overexpressed a **NLS (nuclear localization signal)-tagged** actin variant (R62D) that is impaired in filament formation. To our knowledge, this is currently the only method to perturb nuclear actin filament formation without affecting the cytoplasmic pool of F-actin.

(II) We applied drugs that interfere with actin polymerization globally (Latrunculin B) or via ARP2/3 complex-nucleated F-actin specifically (CK-666).

In both cases, we did not observe any effect on fork stability upon F-actin inhibition (Figure for reviewer #1). These results could be interpreted as the consequence of an actin-independent function of myosin VI in fork stabilization. Alternatively, however, they could result from a myosin VI-independent function of actin upstream of myosin VI's point of action. We consider the latter to be more likely, based on recent findings from the lab of Massimo Lopes, who observed a contribution of F-actin to replication fork reversal, which is upstream of our postulated myosin VI function. A manuscript

showing these results from the Lopes lab is currently under review, but since the data are not published yet, we would prefer not to discuss them or show our actin-related dataset in our study.

Figure for reviewer #1: Perturbation of actin filament formation has no effect on replication fork stability

Ia top: Schematic representation of fiber assay conditions. Bottom: Myosin VI GFP-tail *wildtype* (WT) and the polymerization deficient mutant NLS-actin_R62D were overexpressed in U2OS cells for 48 h, followed by fiber assays performed as shown on top. Combined data from at least 3 independent replicates are shown. IdU/CldU ratios are shown as dot plots with median values. Significance levels were calculated using the Mann-Whitney test from at least 100 fibers per sample (ns: not significant, ****: $p < 0.0001$). Ib: western blots are shown using the indicated antibodies.

IIa top: Schematic representation of fiber assay conditions as indicated. for 72 h, Bottom: U2OS cells were transfected with siRNAs as indicated for 72 h, followed by fiber assays in the presence or absence of LatB or CK666 as indicated. IdU/CldU ratios are shown as dot plots with median values. Significance levels were calculated using the Mann-Whitney test from at least 100 fibers per sample (ns: not significant, ****: $p < 0.0001$). IIb: western blots are shown using the indicated antibodies.

(b) What happens to the actin and myosin VI nuclear levels during replication stress?

Visualization of nuclear actin/F-actin and myosin VI is technically extremely challenging (reviewed in Hurst et al. 2019). To address the reviewer's question, we used confocal microscopy. From a Z-stack, we selected a central nuclear plane and used Alexa Fluor 647-labelled Phalloidin to stain filamentous actin. Unfortunately, our myosin VI-specific antibodies were not suitable for the microscopic approach due to a relatively high nuclear background signal. We therefore chose transient overexpression of GFP-Myosin VI.

Our data show an increase in nuclear F-actin levels upon replication stress, while myosin VI levels appear to remain unchanged (Fig. S3).

Hurst, V., Shimada, K. & Gasser, S.M. Nuclear Actin and Actin-Binding Proteins in DNA Repair. *Trends Cell Biol* **29**, 462-476 (2019).

(c) Can you express the recombinant WRNIP to determine if there is a direct MVI interaction?

Following the reviewer's request, we have performed interaction studies with recombinant proteins and in this manner demonstrate that the interaction between WRNIP1 and Myosin VI is direct (Fig. 3d), but in parts also mediated by ubiquitin conjugates (Fig. 3e, see below).

(d) Is the interaction with WRNIP mediated via Ub? Can you manipulate this process to conclude how the proteins interact?

In GST-pulldown assays with recombinant proteins, we found an enhancement of the interaction between Myosin VI and WRNIP1 by the addition of *in vitro*-generated K63-linked ubiquitin chains (Fig. 3e).

To manipulate the ubiquitin-mediated aspect of the interaction, we utilized the catalytic core of USP2 to disassemble endogenous ubiquitin chains in cellular extracts. Using GST-pulldown assays with these extracts, we observed a decrease in the Myosin VI-WRNIP1 interaction after USP2-treatment (Fig. 3f), suggesting that ubiquitin chains, as suggested by the reviewer, also contribute to this interaction.

(e) You strongly hint towards a motility-based role of myosin VI. I would tone down some of these statements because there is not experimental data to support this conclusion.

We thank the reviewer for this comment and re-formulated our statements throughout the text accordingly.

Furthermore, we eliminated the following statement from our discussion:

“A scenario where myosin VI acts by mediating the transport of the fork protection factor WRNIP1 to its sites of action is consistent with our protein interaction and localization data (Fig. 3c, 5c).”

Reviewer #2

I was specifically asked to look into the mass spectrometry aspects of the paper. The methodology is sound. SILAC was used and 4 GST pull downs repeats were performed including reverse labelling. The methods section is thorough.

We thank this reviewer for this very positive assessment of the mass spectrometry part of our study.

There are nonetheless a few minor elements to provide in the methods:

-please specify if dialysed serum was used in SILAC labeling of HeLas (line 234)

We have made the description of SILAC labeling more specific. The section now reads:

“For SILAC labeling, HeLa cells were cultured for at least 5 passages in SILAC DMEM (Invitrogen) supplemented with dialyzed FBS (Invitrogen) and containing either L-arginine and L-lysine (Merck) or L-arginine [13C6] and L-lysine [2H4] (Cambridge Isotope Laboratories).”

-please specify the trypsin concentration used (line 297)

We have now specified the amount of trypsin used per fraction:

“Proteins were stained using the Colloidal Blue Staining Kit (Life Technologies) and digested in-gel using 0.6 µg of MS-approved trypsin (Serva) per gel fraction.”

-please specify how the 4 repeats were compared and which hits were kept in the final list, i.e. specify which hits were common to all repeats or if less, how many?

The hits used in statistical analysis were those that could be quantified in at least two replicates. In Supplementary Table 1, we now include a column called “Identified in x replicates”, as well as a column displaying the number of unique peptides quantified per protein group.

Out of 490 proteins enriched in the MyUb-GST pulldown, 265 were quantified in four replicates, 75 were quantified in three replicates, and 150 were quantified in two replicates (Fig. S1b). Of note, WRNIP1 was quantified in three replicates.

Reviewer #3

The authors have studied the role of myosin VI in chromosome replication. They provide evidence that i) Myosin VI interacts with replisome components ; ii) Myosin VI depletion affects fork elongation and results into fork processing ; iii) Myosin VI cooperates with WRNIP1 at forks.
General comments

In general, the finding that nuclear Myosin VI influences chromosome replication is original and potentially relevant. Most of the results are convincing.

We thank this reviewer for this very positive evaluation of our work.

However, there are serious issues that must be addressed.

Specific comments

1. The field of reversed forks is already confusing and controversial. The authors abuse of the term reversed fork without providing any direct evidence that forks indeed undergo reversal. The authors have two options: a) if they really want to push the fork reversal concept, then they need to analyze the replication intermediates by EM. b) Remove any statement related to fork reversal from abstract and results sections (including the schemes in the figures). They can eventually speculate about fork reversal in the discussion without being too dogmatic in their conclusions.

We agree with the reviewer that visualization of reversed fork structures via electron microscopy is a powerful technique to study important aspects of the fork reversal process and provides more direct evidence for the phenomenon. However, this assay is established in very few laboratories in the world, and due to the limited capacity of these labs we have been unable so far to engage in a collaboration to directly monitor the effects of myosin VI on fork reversal by EM.

At the same time, our data showing a role of myosin VI in fork stabilization do not rely on the concept of reversed forks. All our results would still be valid if the actual structure protected by myosin VI had a different geometry, such as – for example – a template switching intermediate. Such alternative structure would still need to be brought about by Rad51, HLF, ZRANB3 and SMARCAL1 and would need to be subject to degradation by MRE11 and SLX4/DNA2. These properties have been associated with reversed forks by a number of labs who have established that forks reverse under the conditions we apply in our assays. Thus, without claiming to provide evidence for fork reversal ourselves, we would prefer to build our model on the concept of reversed forks, thus following established procedures in the literature (see references 1-5 below for recent studies centered on fork reversal that do not contain electron microscopy data).

To make this issue transparent, we now include a statement to that effect in our discussion section:

“Based on our placement of myosin VI activity downstream of a series of factors known to mediate fork reversal, such as RAD51, SMARCAL1, HLF and ZRANB3³⁻⁵, we postulate that myosin VI specifically acts on reversed forks; however, our data do not exclude an alternative fork geometry generated by the above remodelers.”

1. Malacaria, E. *et al.* Rad52 prevents excessive replication fork reversal and protects from nascent strand degradation. *Nat Commun* **10**, 1412 (2019).
2. Griffin, W.C. *et al.* A multi-functional role for the MCM8/9 helicase complex in maintaining fork integrity during replication stress. *Nat Commun* **13**, 5090 (2022).
3. Marie, L. & Symington, L.S. Mechanism for inverted-repeat recombination induced by a replication fork barrier. *Nat Commun* **13**, 32 (2022).
4. Paniagua, I. *et al.* MAD2L2 promotes replication fork protection and recovery in a shieldin-independent and REV3L-dependent manner. *Nat Commun* **13**, 5167 (2022).

5. Schleicher, E.M. *et al.* The TIP60-ATM axis regulates replication fork stability in BRCA-deficient cells. *Oncogenesis* **11**, 33 (2022).

2. It would be important to test whether the impact on replication caused by Myosin VI depletion is due to an in trans effect resulting from ATR activation. In other words, does Myosin VI depletion activate ATR? And if yes, does ATR activation play any role in fork stalling/slowing down following Myosin VI depletion?

We thank the reviewer for this valuable suggestion. We performed western blot analyses monitoring CHK1 phosphorylation on S317 and S345 as a proxy for ATR activation in control and myosin VI knockdown conditions. In both cases, a treatment with 4 mM Hydroxyurea for 1 h was used as positive control for ATR activation (Fig. S2b). We did not observe significant differences between myosin VI knockdown and control conditions, suggesting that ATR activation as a consequence of myosin VI-depletion is not the cause of the effects we observe.

3. Figure 4: I do not find the result convincing. This part could be removed.

We disagree with the reviewer. The lack of damage-induced accumulation of WRNIP1 upon myosin VI depletion that we show here is significant and reproducible and adds important mechanistic information to our findings. We would therefore prefer to keep the figure (now Fig. 5).

Reviewer #4

In this manuscript, the authors show the role of nuclear myosin VI in the stabilisation of replication forks. To do that, they identified nuclear interactors of myosin VI by mass spectrometry analysis. They also developed DARPin-specific binders to manipulate the localisation of myosin VI and to confirm the involvement of nuclear myosin VI in controlling the fork stability. This is an original study using different methods to show what is the role of myosin VI on replication forks' stability.

We thank this reviewer for this very positive evaluation of our work.

As requested, I particularly focused on the experiment involving DARPins screening and selection to obtain the myosin VI binders.

1) Concerning the selection of DARPins by ribosome display, the authors use "undesired myosin VI isoforms 2 and 3" as competitors, but no data is shown to actually demonstrate the specificity of the selected DARPins towards isoform 1. The authors should provide data to show the isoform 1 specificity of M6G4.

We apologize for the misunderstanding regarding different myosin VI isoform that, as we now realize, was caused by insufficient explanation. The DARPin we are using in our studies, M6G4, originates from a screen against isoform 1 but actually does not show any isoform specificity (Fig. S4c). Furthermore, the cellular system that we are using (U2OS) in our functional studies exclusively expresses isoform 2 (the short isoform). We therefore do not discuss the different isoforms in the manuscript to avoid unnecessary complications. We made changes in the manuscript to clarify this issue.

2) The authors should clearly indicate in the selection protocol by ribosome display what are the constructs employed for myosin VI isoforms 2 and 3 (full length proteins or truncated proteins? which amino acids were used?).

For DARPin selection (isoform 1) and “counter-selection” (isoforms 2+3) we used truncated versions of myosin VI isoforms comprising the MIU and MyUb domains (aa 992-1122). The precise construct borders are now mentioned in the manuscript.

3) Throughout the entire manuscript, the authors mention myosin VI, however there are different isoforms and the DARPins are theoretically isoform 1 specific. Therefore, for clarity purposes, the authors should indicate in the introduction the differences between the three isoforms (localisation, expression, function?) and explain why they selected DARPins against the isoform 1 only.

We apologize for our sloppiness regarding myosin VI isoforms in the initial manuscript. As explained above in points 1) and 2), we hope that the changes we made in the current version of our manuscript will help to avoid any isoform-related confusions.

4) While the strategy used by the authors to screen their binders after ribosome display is interesting (HTRF), no data is available to show this. Therefore, the authors should add the data of the HTRF screening in a figure and explain these results within the results section (l139: “we obtained 54 candidates from the library...”).

A figure showing the HTRF dataset was added and extended explanations in the results section and in the respective figure legend are given (Fig. S4a).

5) The authors should determine the Kd of their lead DARPin (M6G4) to confirm the panning strategy used by the authors “l430: increasing washing steps to enrich for binders with slow off-rates and thus high affinities”.

We used Surface Plasmon Resonance to estimate the Kd of our lead DARPin, M6G4 (Fig S4e).

6) The authors should indicate in a supplementary file, the DNA and protein sequences of the anti-myosin VI DARPin M6G4.

The requested information is now supplied in a supplementary file.

7) The authors should show whether the binding of DARPin M6G4 affects myosin VI PPIs such as WRNIP1 (if any inhibition, it would probably be by steric hindrance rather than direct PPI inhibition). This is an important question to answer before using the DARPin M6G4 as relocalisation tool of myosin VI.

The requested experiment was performed as suggested. We found that M6G4 has no effect on the ability of myosin VI to bind to WRNIP1 and ubiquitin (Fig. S4f). In addition, our observation that overexpression of the NLS-tagged M6G4 DARPin does not interfere with the protective function of myosin VI during replication stress (Fig. 4e), provides further evidence against a negative effect of our lead DARPin on the interaction of myosin VI with WRNIP1 or ubiquitin.”

Comments to the authors unrelated to the DARPin screening and selection:

8) Abstract: l27-28: “Using nuclear localization sequence (NLS) and ubiquitin E3-fusion DARPins to manipulate myosin VI levels in a compartment-specific manner”. This sentence sounds like the targeted degradation of myosin VI by DARPin-E3 ligase fusion is compartment-specific, which is not the case here. This sentence should be rephrased.

We thank the reviewer for this comment. We have now re-phrased this sentence accordingly.

9) Fig3b: a proteasome inhibitor control would be required to show that the degradation of myosin VI by the DARPin-RINGx2 construct is proteasome dependent.

Using the proteasome inhibitor MG-132, we were able to show that doxycycline-induced myosin VI degradation in the 2R8 cell line is dependent on proteasome function (Fig. S4h).

10) L158-161: the authors should directly test the NES-DARPin M6G4 construct to exclude the endogenous myosin VI from the nucleus. Even if this experiment does not work (with the NES-DARPin M6G4), it is important to show it rather than a test case with a GFP and an anti-GFP DARPin. The argument stating that the NES-DARPin fusion is synthesised in the cytosol is correct but the GFP (or myosin VI) are also synthesised in the cytosol, so the DARPins should be able to capture the GFP/myosin VI before it goes into the nucleus.

As suggested by the reviewer, we have attempted to extract myosin VI from the nucleus with an NES-tagged DARPin. To this end, we generated the respective NES-M6G4 and NES-E3_5 constructs. Unfortunately, the NES-M6G4 construct was expressed at very low levels within cells (see Figure for reviewer #4), which precluded a meaningful interpretation of the experiment. Furthermore, due to the low nuclear abundance of myosin VI itself, it is difficult to quantify its nuclear levels in a reliable manner. It would therefore be close to impossible to verify a successful extraction of myosin VI from the nucleus via an NES-DARPin construct. Our attempts have therefore remained inconclusive. Since we have shown the importance of the nuclear pool of myosin VI for the protection of stalled forks by other means (Fig. 4d-f), we decided to remove the NES-GFP-DARPin statement that was criticized by reviewer #4.

NES-mRuby

Figure for reviewer #4. The NES-mRuby-tagged M6G4 DARPin is poorly expressed.

U2OS cells were transfected with NES-DARPin constructs as indicated. 24 h post-transfection, cells were lysed and subjected to western blot analysis with an mRuby-specific antibody. Ponceau staining was performed to show equal loading.

REVIEWERS' COMMENTS

Reviewer #1 (Remarks to the Author):

The authors have addressed my comments and I look forward to seeing the manuscript published as soon as possible. This is a highly interesting and innovative story which greatly expands the nuclear myosin field.

Reviewer #2 (Remarks to the Author):

I am satisfied with the answers that the authors have provided and how the manuscript was amended.

Reviewer #3 (Remarks to the Author):

The authors have addressed my concerns

Reviewer #4 (Remarks to the Author):

The authors replied to all my comments using appropriate experiments. I would like to thank the authors for this nice work.